# Network compression using Correlation Analysis of Layer Responses

## Abstract

Principal Filter Analysis (PFA) is an easy to implement, yet effective method for neural network compression. PFA exploits the intrinsic correlation between filter responses within network layers to recommend a smaller network footprint. We propose two compression algorithms: the first allows a user to specify the proportion of the original spectral energy that should be preserved in each layer after compression, while the second is a heuristic that leads to a parameter-free approach that automatically selects the compression used at each layer. Both algorithms are evaluated against several architectures and datasets, and we show considerable compression rates without compromising accuracy, e.g., for VGG-16 on CIFAR-10, CIFAR-100 and ImageNet, PFA achieves a compression rate of 8x, 3x, and 1.4x with an accuracy *gain* of 0.4%, 1.4% points, and 2.4% respectively. In our tests we also demonstrate that networks compressed with PFA achieve an accuracy that is very close to the empirical upper bound for a given compression ratio. Finally, we show how PFA is an effective tool for simultaneous compression and domain adaptation.

## 1 Introduction

Despite decades of research, the design of neural networks is still an empirical process. Practitioners make design choices, such as the number of layers, type of layers, number of filters per layer, etc., based on intuition or brute-force search. Nevertheless, the performance of these algorithms, together with the advances of GPU devices, have led to a growing popularity of these techniques in both academia and industry. Recent advances are unveiling some properties of neural networks. For example, there is a consensus that depth can accelerate learning, and that wider layers help optimization (Arora et al., 2018; Livni et al., 2014; Cohen et al., 2018). However, in practical applications, the size of these networks is often a limiting factor when deploying on devices with constrained storage, memory, and computation resources.

Another known neural network property is that the responses of a layer exhibit considerable correlation (Denil et al., 2013), inspiring the idea of learning decorrelated filters (Cogswell et al., 2016; Rodríguez et al., 2017). These algorithms propose a modified loss function to encourage decorrelation during training and show that accuracy improves with decorrelated filters. However, such algorithms focus on training and do not address network compression. *Our hypothesis is that layers that exhibit high correlation in **filter responses** could learn equally well using a smaller number of filters.*

Principal Filter Analysis (PFA) draws from the recent findings that it is easier to start with an over-parametrized network and it then exploits intra-layer correlation for guiding network compression. PFA analyzes a trained network and is agnostic to the training methodology and the loss function. Inference is performed on a dataset, and the correlation within the responses of each layer is used to provide a compression recipe. A new smaller architecture based on this recipe can then be retrained.

We propose two closed-form algorithms based on spectral energy analysis for suggesting the number of filters to remove in a layer:

**PFA-En** uses Principal Component Analysis (PCA) (Hotelling, 1933) to allow a user to specify the proportion of the energy in the original response that should be preserved in each layer;

**PFA-KL** is a heuristic that leads to a parameter-free approach that uses Kullback-Leibler (KL) divergence (Kullback & Leibler, 1951) to identify the number of redundant filters.

Based on the new suggested number of filters per layer identified by PFA, we remove those that are maximally correlated with other filters and fine-tune the network by retraining. Both PFA algorithms are straightforward to implement and, as shown in Sec. 4, they achieve better compression and, in most cases, better accuracy than the state of the art on several datasets and architectures. In Sec. 4 we also show how PFA can be used to perform simulations compression and domain adaptation.

## 2 RELATED WORK

The field of network compression encompasses a wide range of techniques that can be grouped into the following families: quantization, knowledge distillation, tensor factorization and network pruning.

**Quantization algorithms** compress networks by reducing the number of bits used to represent each weight (Wu et al., 2016; Han et al., 2016a; Rastegari et al., 2016; He & Cheng, 2018; Wu et al., 2018).

**Knowledge distillation** (Hinton et al., 2014) and **model compression** (Buciluǎ et al., 2006) aim to create a simpler model that mimics the output of a more complex model. Variations on this concept include Ba & Caurana (2014); Romero et al. (2015); Chen et al. (2016).

**Tensor factorization** algorithms exploit the redundancy present in convolution layers by replacing the original tensors with a sequence of smaller or sparser counterparts that produce similar responses (Denton et al., 2014; Lebedev et al., 2014; Jaderberg et al., 2014; Cheng et al., 2016; Masana et al., 2017; Wen et al., 2017; Yu et al., 2017; Alvarez & Salzmann, 2017; Peng et al., 2018).

**Network pruning** is a family of techniques that compress networks by iteratively removing connections based on the salience of their weights. Early work, like Optimal Brain Damage (LeCun et al., 1990) and Optimal Brain Surgeon (Hassibi et al., 1993), targeted fully connected networks. Recent work can be divided into two sub-families: sparse pruning (Han et al., 2016b; Srinivas & Babu, 2015; Wen et al., 2016; Tung & Mori, 2018; Aghasi et al., 2017; Carreira-Perpinan & Idelbayev, 2018; Zhang et al., 2018; Dai et al., 2018), where individual neurons are removed, and structured pruning (Li et al., 2017; He et al., 2017; Luo et al., 2017; Molchanov et al., 2017; Yu et al., 2018), where entire filters are removed.

PFA falls within the family of structured network pruning. Some of these techniques, e.g., Li et al. (2017), require user defined parameters that are hard to choose and whose effect on the footprint is difficult to predict (see Sec. 4.6 for a more detailed discussion). Others also require modification of the loss function, e.g., Liu et al. (2017). In contrast, PFA-En has only one intuitive parameter, which is the proportion of the response energy to be preserved at each layer, and PFA-KL is parameter-free. Furthermore, instead of learning the saliency of the filters during training by modifying the loss function, PFA estimates it after training without requiring knowledge of the training details. This makes PFA applicable to any trained network, without the need to know its loss function.

Within the structured pruning family, there are approaches based on the singular value decomposition (SVD) (Xue et al., 2013; Nakkiran et al., 2015; Prabhavalkar et al., 2016), where a new set of filters are obtained by projecting the original weights onto a lower dimensional space. PFA differs from these methods by the fact that SVD is performed on the responses of the layers rather than on the filter weights, and no projection is done. This is particularly relevant for domain adaption applications, where a trained network is specialized for a different task. Techniques that make compression decisions based on *weights*, rather than the *responses*, cannot take into account the specificity of the task. As shown in Sec. 4.7, PFA derives different architectures from the same initial model when analyzed for different tasks.

Some methods, e.g., Li et al. (2017); Masana et al. (2017); Peng et al. (2018), also reason on the layer responses. These techniques aim to find a smaller set of filters that, in different ways, minimize the reconstruction error of the feature maps or the response output. Note that PFA uses the spectral energy of the filters' responses only to decide how many filters should be preserved. Similarly, the subsequent filter selection process does not take the reconstruction error into account. PFA is also philosophically and practically different: PFA uses the concept of correlation within the responses to identify redundancy within a layer. In practice this means that PFA can compress all layers in one run, while the majority of the techniques that use responses need to operate on one layer at the time.

Finally, PFA is orthogonal to the quantization, tensor factorization and distillation methods, and could be used as a complementary strategy to further compress neural networks.

# 3 PRINCIPAL FILTER ANALYSIS

In this section, PFA-En and PFA-KL algorithms are described in detail. Both algorithms share the idea of exploiting correlations between responses in convolutional layers and neurons in fully connected layers to obtain a principled recommendation for network compression.

## 3.1 DEFINITIONS

PFA is inherently data driven and thus takes advantage of a dataset $\{\mathbf{X}_i\} \in \mathbb{R}^{M \times I}$ where $\mathbf{X}_i$ is the $i^{th}$ input data sample, $M$ is the number of samples in the dataset, and $I$ is the input dimensionality. Typically, this dataset is the data used to train the network, but it can also be a representative set that covers the distribution of inputs likely to be encountered. Without loss of generality, we assume that the input data are images:$\{\mathbf{X}_i\} \in \mathbb{R}^{M \times H \times W \times C}$, where $H$ is the image height, $W$ is the image width and $C$ is the number channels.

Let $\mathbf{T}_i^{[\ell]} \in \mathbb{R}^{1 \times H^{[\ell]} \times W^{[\ell]} \times C^{[\ell]}}$ be the output tensor produced by a given layer $\ell$ of a network on the $i^{th}$ input sample. Any operation in the network is considered a layer (e.g., batch normalization, ReLU, etc.). In this work we analyze the output of convolutional and fully connected layers. However, PFA can be used to analyze the output of any layer in the network. Recurrent neural networks could also be analyzed, but are out of the scope for this paper.

For a convolutional layer $\ell$, let $\mathbf{W}^{[\ell]} \in \mathbb{R}^{f_h^{[\ell]} \times f_w^{[\ell]} \times C^{[\ell-1]} \times C^{[\ell]}}$ be the set of $C^{[\ell]}$ trainable filters with a kernel of size $f_h^{[\ell]} \times f_w^{[\ell]} \times C^{[\ell-1]}$. Therefore, we can formally express $\mathbf{T}_i^{[\ell]}$ produced by the convolutional layer as $\mathbf{T}_i^{[\ell]} = \mathbf{W}^{[\ell]} * \mathbf{T}_i^{[\ell-1]}$ with $\mathbf{T}_i^{[0]} = \mathbf{X}_i$, where $*$ denotes the convolution operator. We omit the bias term to improve readability.

We define the response vector $\mathbf{a}_i^{[\ell]} \in \mathbb{R}^{C^{[\ell]}}$ of a given layer $\ell$ with respect to an input $\mathbf{X}_i$ to be the spatially max-pooled and flattened tensor $\mathbf{T}_i^{[\ell]}$ (i.e., max-pooling over the dimensions $H^{[\ell]}$ and $W^{[\ell]}$). For fully-connected layers, $\mathbf{W}^{[\ell]} \in \mathbb{R}^{C^{[\ell]}}$, with $C^{[\ell]}$ being the number of neurons in layer $\ell$. The output tensor is $\mathbf{T}_i^{[\ell]} = \mathbf{W}^{[\ell]} \mathbf{T}_i^{[\ell-1]}$, and since no pooling is required, the response vector is $\mathbf{a}_i^{[\ell]} = \mathbf{T}_i^{[\ell]} \in \mathbb{R}^{C^{[\ell]}}$.

Let $\mathbf{A}^{[\ell]} = [\mathbf{a}_1^{[\ell]}, \ldots, \mathbf{a}_M^{[\ell]}]^\top \in \mathbb{R}^{M \times C^{[\ell]}}$ be the matrix of responses of layer $\ell$ given a dataset with $M$ samples.

Lastly, let $\boldsymbol{\lambda}^{[\ell]} \in \mathbb{R}^{C^{[\ell]}}$ be the distribution of eigenvalues of the covariance matrix of $\mathbf{A}^{[\ell]}$ *sorted in descending order and normalized to sum to 1*. In the following sections we present two algorithms that exploit the distribution $\boldsymbol{\lambda}^{[\ell]}$ to guide network compression.

## 3.2 COMPRESSION RECIPES

The distribution $\boldsymbol{\lambda}^{[\ell]}$ provides insight into the correlation within layer $\ell$. The closer $\boldsymbol{\lambda}^{[\ell]}$ is to a uniform distribution, the more decorrelated the response of the filters and the more uniform their contribution to the overall response energy. Conversely, the closer $\boldsymbol{\lambda}^{[\ell]}$ is to a Dirac $\delta$-distribution, the more correlated the filters. Our hypothesis is that layers that exhibit high correlation in filter responses could learn equally well using a smaller number of filters.

Now that we have defined the key ingredient ($\boldsymbol{\lambda}^{[\ell]}$) for PFA, we present two strategies that produce a compression recipe with the goal of maximizing compression while reducing correlation. Let a compression recipe $\boldsymbol{\Gamma} = \{\gamma^{[\ell]}\}$, with $\gamma^{[\ell]} \in (0, 1]$, be the set of compression factors applied to each of the $L$ layers included in the analysis. For example, $\gamma^{[3]} = 0.6$ means that we keep 60% of the filters in layer 3.

Once the correct number of filters have been determined by the recipe, one could further proceed to choose which filters should be kept. We call this *filter selection* and we outline it in Sec. 3.2.3.

### 3.2.1 PFA-EN: ENERGY-BASED RECIPE

PCA can be used for dimensionality reduction by performing a linear mapping to a lower dimensional space that maximizes the variance of the data in this space. This can be accomplished by extracting the eigenvectors and eigenvalues of the covariance matrix. The original data are then reconstructed using the minimum number of eigenvectors that correspond to the eigenvalues that sum up to the desired energy factor $\tau$. Inspired by this strategy, we propose to keep the minimum set of filters such that a fraction of response energy greater or equal to a user defined $\tau$ is preserved. We define the energy at a given compression ratio for a layer as

$$\mathcal{E}(\gamma^{[\ell]}) = \sum_{k=1}^{\lceil \gamma^{[\ell]} \cdot C^{[\ell]} \rceil} \lambda_k^{[\ell]}, \tag{1}$$

and we propose to re-architect the network according to the following recipe:

$$\mathbf{\Gamma}_{\mathcal{E}}^{\star}(\tau) = \{\min \gamma^{[\ell]}\} \quad \text{s.t.} \quad \mathcal{E}(\gamma^{[\ell]}) \geq \tau, \quad \forall \ell. \tag{2}$$

The parameter $\tau$ provides the user with the ability to guide the compression ratio.

PFA-En has the advantage of being tightly connected to well-established dimensionality reduction techniques based on PCA, it is simple to implement and uses a single, highly intuitive parameter. Furthermore, since evaluating the size of a model (or its FLOPs) obtained at different energy thresholds is easy and fast, it is straightforward to replace the parameter $\tau$ with the desired footprint (or FLOPs) after compression. Being able to specify a target footprint instead of an energy threshold gives PFA-En even more appeal for practical use cases.

### 3.2.2 PFA-KL: KL DIVERGENCE-BASED RECIPE

We propose an alternative formulation to obtain an optimal recipe $\mathbf{\Gamma}_{\text{KL}}^{\star}$, based on KL divergence. This formulation is a heuristic that frees PFA from the use of any parameter. As previously mentioned, a distribution $\boldsymbol{\lambda}^{[\ell]}$ similar to a flat distribution implies an uncorrelated response of the filters in layer $\ell$. Therefore, the farther the distribution $\boldsymbol{\lambda}^{[\ell]}$ is from a flat distribution the more layer $\ell$ can be compressed.

Let us define $\boldsymbol{u}$ as the *desired* uniform distribution (no correlation between filters), and $\boldsymbol{d} = \text{Dirac}(k)$ as the *worst case* distribution (all filters are perfectly correlated). We can measure the dissimilarity of the actual distribution, $\boldsymbol{\lambda}^{[\ell]}$, from the *desired* distribution, $\boldsymbol{u}$, as the KL divergence $\text{KL}(\boldsymbol{\lambda}^{[\ell]}, \boldsymbol{u})$. The upper bound of which is given by $u_{\text{KL}} = \text{KL}(\boldsymbol{d}, \boldsymbol{u})$, while the lower bound is 0. Notice that one could replace the KL divergence with any dissimalarity measure between distributions, such as $\chi^2$ or the earth mover's distance (Rubner et al., 1998).

Intuitively, when the actual distribution is identical to the ideal distribution (i.e., no correlation found) then we would like to preserve all filters. On the other hand, when the actual distribution is identical to the worst case distribution (i.e., all filters are maximally correlated) then one single filter would be sufficient. The proposed KL divergence-based recipe is a mapping $\psi : [0, u_{\text{KL}}] \mapsto (0, 1]$; a divergence close to the upper bound results in a strong compression and a divergence close to the lower bound results in a milder compression:

$$\mathbf{\Gamma}_{\text{KL}}^{\star} = \left\{ \psi \big( \text{KL}(\boldsymbol{\lambda}^{[\ell]}, \boldsymbol{u}), u_{\text{KL}} \big) \right\}, \quad \forall \ell. \tag{3}$$

In this work, we use a simple linear mapping $\psi(x, u_{KL}) = 1 - x/u_{KL}$. Other mappings were explored, leading to different degrees of compression; however, we have empirically observed that a linear mapping produces good results that generalize well across networks.

### 3.2.3 FILTER SELECTION

The recipes produced by PFA-En and PFA-KL provide the number of filters, $F^{[\ell]} = \lceil \gamma^{[\ell]} \cdot C^{[\ell]} \rceil$, that should be kept in each layer, but do not indicate which filters should be kept. One option is to retrain the compressed network from a random initialization (*from scratch*). In this case it does not matter which filters are chosen. An alternative is to select which filters to keep and use their values for initialization. We do this by removing those filters in each layer that are maximally correlated.

We compute the $\ell_1$-norm of each row of the correlation matrix from the matrix $\mathbf{A}^{[\ell]}$ and remove the filter with the largest norm. If more filters need to be chosen then we update the correlation matrix by removing the previously selected filter, and iterate until the desired number of filters has been removed. In the rare, but theoretically possible, case in which two filters have the same $\ell_1$-norm we choose the one with the highest individual correlation coefficient.

## 4 EXPERIMENTS

To evaluate PFA, we analyze several network architectures and datasets, and our results are compared to the state of the art. Specifically we will contrast PFA against the filter pruning approach (FP) in Li et al. (2017), the network slimming approach (NS) in Liu et al. (2017), the variational information bottleneck approach (VIB) in Dai et al. (2018) and the filter group approximation approach (FGA) in Peng et al. (2018). For comparison, we focus on the compression ratio and the accuracy change, measured in percentage points (pp), obtained by the compressed architectures. This enables plotting various techniques in the same plot, even if the accuracy of each original architecture is slightly different because of different training strategies used.

After the Sate of the Art quantitative comparison, PFA is tested on the task of simultaneous compression and domain adaptation.

### 4.1 NETWORKS AND DATASETS

Three networks are evaluated in this paper: VGG-16 (Simonyan & Zisserman, 2015) (we use the version proposed by Zagoruyko (2015) for CIFAR), ResNet-56 (He et al., 2016), and SimpleCNN, a small CNN we use to demonstrate the capabilities of PFA. Refer to Appendix A for the architecture details. We train and test all the networks on CIFAR-10 and CIFAR-100 (Krizhevsky, 2009). In addition we test VGG-16 (Long et al., 2015) on ImageNet (Deng et al., 2009).

### 4.2 METHODOLOGY

A baseline for each architecture is obtained by training using 10 random initializations – we choose the initialization that leads to the highest test accuracy and perform inference on the training set to obtain the responses at each layer ($\mathbf{A}^{[\ell]}$) needed for the PFA analysis. PFA analyzes all layers in parallel to obtain its recipes. Compressed architectures are created by following the recipes from each PFA strategy. For each new compressed architecture, we perform two types of training. First, we retrain with 10 different random initializations and this set of results are referred to as *Scratch*. Second, we retrain 10 times using the filter selection method described in Sec. 3.2.3, and fine-tune the compressed network starting from the weights of the selected filters. The accuracy reported is the mean of these 10 retrainings. Note that the retraining is done without hyper-parameter tuning. While this is a sub-optimal strategy, it removes any ambiguity on how well the parameters were tuned for the baseline compared to the compressed networks. In practice, one would expect to attain even better results if parameter sweeping was done on the compressed networks.

In all experiments we use the SGD optimizer with no decay, learning rate 0.1, Nesterov momentum 0.9, 50 epochs for SimpleCNN and 160 for VGG-16 and ResNet-56.

PFA-En is computed for every energy value: $\tau \in \{0.8, 0.85, 0.9, 0.925, 0.95, 0.96, 0.97, 0.98, 0.99\}$, whereas PFA-KL is parameter-free and is computed once per baseline network.

### 4.3 EMPIRICAL UPPER-BOUND FOR SIMPLECNN ON CIFAR-10 AND CIFAR-100

An empirical upper bound on the accuracy at different compression ratios is obtained by randomly choosing how many filters to remove at each layer. By repeating this a sufficient number of times, the best result at each compression ratio can be considered an empirical upper bound for that architecture and footprint, labeled *Upper bound*. On the other hand, the result averaged across all trials is representative of how easy (or difficult) it is to randomly compress a network without hurting its accuracy, labeled *Avg. random*. Results tagged with the word *Scratch* were trained from a random initialization while those without the tag were obtained using filter selection. In these experiments we generated and trained 300 randomly pruned architectures for each compression ratio. In Fig. 1 the

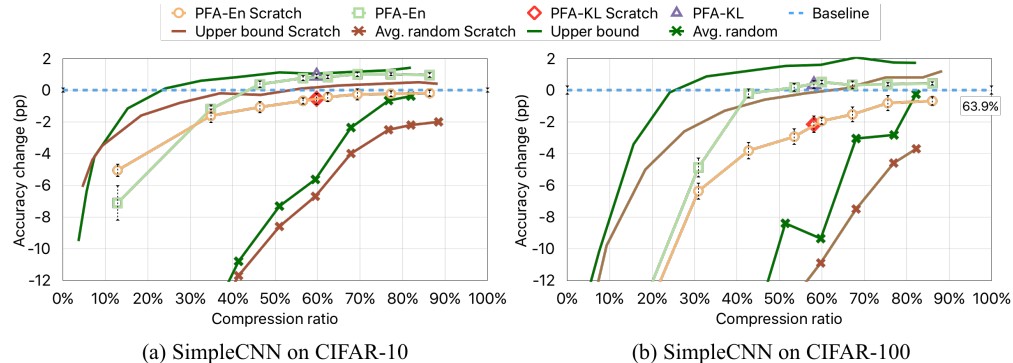

(a) SimpleCNN on CIFAR-10        (b) SimpleCNN on CIFAR-100

Figure 1: Results of different SimpleCNN compressed networks. Accuracy change in the $y$ axis is reported in percentage points (error bars show the standard deviation of multiple runs). Note how all PFA algorithms lie close to the upper bound while the random pruning severely degrades accuracy. Also note how the filter selection strategy improves over the random initalization (*Scratch*).

accuracy drops for *Avg. random* are particularly severe, showing that in general random pruning is not an effective strategy, while *Upper bound* shows excellent results.

On CIFAR-10, both PFA strategies trained from scratch produce results close to *Upper bound Scratch* and significantly better than the *Avg. random Scratch* results. This indicates that choosing the correct number of filters for each layer is crucial even for footprints that are close to the original model, and shows that PFA is able to perform this choice close to optimally. PFA-En *Scratch* and PFA-KL *Scratch* both achieve a footprint close to 60% with an accuracy drop of only 0.5 pp.

When performing filter selection, rather than retraining from scratch, both PFA-En and PFA-KL improve their accuracy by about 1 pp *above* the *Scratch* version for footprints above 40%. For footprints above 42%, the filter selection strategy performs even better than *Upper bound Scratch*, whereas for footprints less than 40%, the accuracy starts to degrade. On CIFAR-100, filter selection increases its gain up to 2 pp compared to the *Scratch* version, despite the more challenging nature of the dataset as indicated by the faster drop in accuracy past the 40% footprint mark.

We have found that the filter selection strategy converges faster during training and performs consistently better across different architectures and datasets, hence from now on we will only report results using PFA with filter selection.

Interestingly, at the 10% footprint mark a random initialization appears to be better than the use of filter selection. It is possible that when keeping an extremely small number of filters, the starting point provided by the filter selection becomes a local minimum that is difficult to escape. For thin layers in relatively small architectures (like SimpleCNN), a random initialization may give more room for exploration during the learning phase.

## 4.4 VGG-16 and ResNet-56 on Cifar-10, Cifar-100 and ImageNet

For VGG-16 we compare the results of PFA with those reported by FP, NS, VIB[1] and FGA[2], after a single iteration of NS for a direct comparison with the other methods (see Fig. 2a, 2b and Tab. 1). Note that VIB performs sparse pruning (single neurons) rather than structured pruning. The main results are summarized in Tab. 1.

PFA-En and PFA-KL outperform all the other methods both on CIFAR-10 and CIFAR-100 in terms of compression ratio. On CIFAR-10, PFA-En ($\tau = 0.98$) obtains a model 2.5x, 2.8x and 4.5x smaller than NS, FP and VIB respectively. On CIFAR-100, PFA-En also obtains models 1.2x, 2x smaller than NS and VIB. Despite achieving smaller models the accuracy change is comparable to that of the other techniques: all techniques achieve an accuracy change between -1pp and +2pp from the baseline. Similarly, the FLOPs reduction is inline with that achieved by other techniques. Note that

---

[1] Error of the original full models kindly provided by the authors of VIB.

[2] Footprint results kindly provided by the authors of FGA.

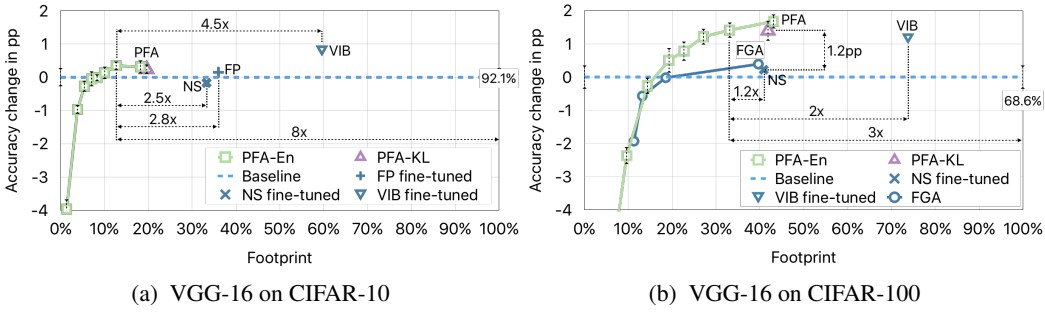

(a) VGG-16 on CIFAR-10  (b) VGG-16 on CIFAR-100

Figure 2: Results for VGG-16 compressed networks. Accuracy change in the $y$ axis is reported in percentage points. Notice how the accuracy obtained by PFA is comparable to the state of the art while achieving much better compression ratios.

Table 1: Results on CIFAR-10, CIFAR-100 and ImageNet for VGG-16 and ResNet-56. PFA-En uses $\tau = 0.98$.

|  |  |  | FP | NS | VIB | FGA | PFA-En | PFA-KL |
|---|---|---|---|---|---|---|---|---|
| **CIFAR-10** | VGG-16 | % Footprint | 36.0% | 33.3% | 59.6% | - | **12.7%** | 19.6% |
|  |  | % FLOPs | 65.8% | 61.4% | **37.1%** | - | 39.1% | 39.4% |
|  |  | Δ Accuracy | 0.15 | -0.17 | **0.81** | - | 0.40 | 0.24 |
|  | ResNet-56 | % Footprint | 86.3% | - | - | - | 61.5% | **59.6%** |
|  |  | % FLOPs | 72.4% | - | - | - | 69.9% | **61.5%** |
|  |  | Δ Accuracy | **0.02** | - | - | - | -0.18 | -0.61 |
| **CIFAR-100** | VGG-16 | % Footprint | - | 40.9% | 73.8% | 39.7% | **33.1%** | 41.9% |
|  |  | % FLOPs | - | 69.1% | 64.9% | 60.0% | 57.1% | **53.9%** |
|  |  | Δ Accuracy | - | 0.22 | 1.17 | 0.39 | **1.40** | **1.40** |
|  | ResNet-56 | % Footprint | - | - | - | - | 81.5% | **73.6%** |
|  |  | % FLOPs | - | - | - | - | 79.4% | **66.7%** |
|  |  | Δ Accuracy | - | - | - | - | **-1.68** | -2.85 |
| **ImageNet** | VGG-16 fully conv. | % Footprint | - | - | - | 74.4% | - | **69.3%** |
|  |  | % FLOPs | - | - | - | **43.0%** | - | 52.7% |
|  |  | Δ Accuracy top-1 | - | - | - | 0.82% | - | **2.39%** |
|  |  | Δ Accuracy top-5 | - | - | - | 0.94% | - | **1.41%** |

PFA-En obtains similar model sizes as FGA on CIFAR-100, but with better accuracy for compression ratios over 13%. At a comparable compression ratio of 40%, PFA-En achieves +1pp better accuracy than FGA.

For ResNet-56 on CIFAR-10 the same conclusions hold true. PFA-En ($\tau = 0.98$) achieves an accuracy similar to FP but with a smaller footprint: 61.5% compared to 86.3%. PFA-KL achieves an even smaller footprint: 59.6%, with an accuracy drop of only 0.6 pp. Similar results are observed on CIFAR-100, as shown in Fig. B.1b in Appendix B.

For VGG-16 on ImageNet we compare PFA-KL with FGA (note that NS also reports results on VGG-16 but with fully connected layers). While FGA reports footprints between 74.4% and 23.18% we report in Tab. 1 the footprint most closely comparable with PFA-KL: at a slightly smaller footprint PFA-KL achieves a better accuracy improvements than FGA, increasing the accuracy over the baseline by 2.4% in the top-1 accuracy.

In summary, independently of the architecture or dataset, PFA consistently provides better compression or accuracy than the state of the art.

### 4.5 ON THE COMPLEXITY AND SCALABILITY OF PFA

The complexity of PFA (excluding the inference step), with respect to number of filters and dataset size, is dominated by the the PCA analysis which, for a given layer, is $O(mn^2 + n^3)$, $n$ being the number of filters, and $m$ the number of samples. For example, for ImageNet (Deng et al., 2009), $m$=1.2M, and assuming a VGG-16 architecture with layers of size $n$ equal to 64, 128, 256, 512, and 4096, the time to compute PFA per layer is roughly 1.24s, 2.8s, 4.6s, 9.3s, and 127.5s respectively (single CPU @ 2.30GHz). The complexity of the filter selection only depends on the layer size. In the worst case is $O(\tilde{n}n^2)$, where $\tilde{n}$ is the number of filters to remove.

Considering that PFA has to run once at the end of the training step, the time consumed by PFA is negligible compared to the whole training time. In exchange for this negligible extra-time, PFA provides the long-term benefit of a smaller footprint and faster inference, which, in the lifetime of a deployed network, including re-training when new data becomes available, will quickly surpass the time initially required by PFA.

### 4.6 LIMITATION OF THE STATE OF THE ART

All the techniques we used for comparison achieve good results in term of maintaining accuracy and reducing FLOPs. In our view their main limitation, and difference with respect to PFA, is in the user defined parameters that they rely upon.

NS has two crucial parameters: the weight of the sparsity regularizer, and the percentage of filters to be pruned. The weight has a direct impact on the induced sparsity, however, there is no intuitive way to set this parameter and it needs to be tuned for each architecture and dataset. In addition, setting the same percentage of filters to be pruned at each layer for the whole network ignores the relative effect of those filters on the accuracy and the footprint of the network.

Setting the thresholds in FP is non-trivial and it requires the user to choose the compression thresholds based on a pre-analysis that provides insight on the sensitivity of each layer to pruning.

Both VIB and FGA require a user tuned parameter that controls how much each layer is compressed. The tuning is different depending on the network-dataset combination and depending on the layer. From the results there seem to be no intuitive way to set this parameter other than trial and error.

In contrast, PFA achieves comparable accuracy and FLOPs with a much higher compression ratio. In addition, one of the advantages of PFA is that it requires a single intuitive parameter (for example the desired footprint in PFA-En), or it is parameter-free (PFA-KL). On the other hand, all the state of the art techniques we used for comparison require a tuning step.

### 4.7 SIMULTANEOUS COMPRESSION AND DOMAIN ADAPTATION USING PFA

The typical PFA use case is to compress a network for the same initial domain $\mathcal{D}_A$ used for training. However, PFA can be computed using data from a different domain, $\mathcal{D}_Z$. The result is a compressed architecture specialized for the target domain $\mathcal{D}_Z$ that takes advantage of the training initially performed on $\mathcal{D}_A$. In this section we show that PFA derives different architectures from the same initial model when analyzed for different tasks, hence, in addition to compression PFA can also be used for domain adaptation.

In order to test PFA on domain adaptation we randomly sample classes out of the original 100 contained in CIFAR-100. We generate two sub-sets ($\mathcal{D}_Z^{R1}, \mathcal{D}_Z^{R2}$) of 10 classes each, and four sub-sets ($\mathcal{D}_Z^{S1}, \mathcal{D}_Z^{S2}, \mathcal{D}_Z^{S3}, \mathcal{D}_Z^{S4}$) of 2 classes each. We perform experiments adapting from initial domains $\mathcal{D}_A^{C10} = $ CIFAR-10 and $\mathcal{D}_A^{C100} = $ CIFAR-100 to the above mentioned target domains. Note that all $\mathcal{D}_Z$ are subsets of $\mathcal{D}_A^{C100}$ but *not* of $\mathcal{D}_A^{C10}$. For each adaptation $\mathcal{D}_A \rightarrow \mathcal{D}_Z$ we run the following experiments using a VGG-16 model:

- **Full scratch**: Train from scratch on domain $\mathcal{D}_Z$.

- **Full fine**: Train from scratch on domain $\mathcal{D}_A$ and fine-tune on $\mathcal{D}_Z$.

- **PFA scratch**: Train from scratch on domain $\mathcal{D}_A$, run PFA-KL on domain $\mathcal{D}_Z$ and train compressed architecture from scratch on $\mathcal{D}_Z$.

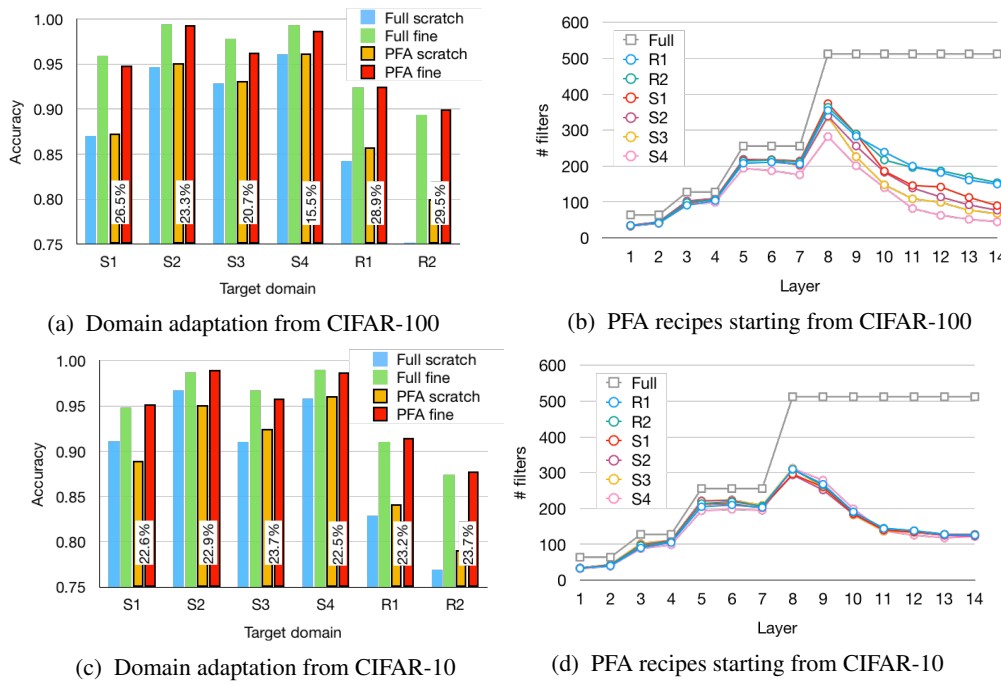

(a) Domain adaptation from CIFAR-100

(b) PFA recipes starting from CIFAR-100

(c) Domain adaptation from CIFAR-10

(d) PFA recipes starting from CIFAR-10

Figure 3: Domain adaption from CIFAR-100 and from CIFAR-10. (a,c) *PFA fine* matches the accuracy of *PFA full* while using architectures more than 4x smaller. *PFA fine* largely outperforms the full model trained from scratch *Full scratch*. The vertical percentage labels show the PFA compression ratio. In (b,d), recipes for VGG-16 trained on CIFAR-100 and CIFAR-10 using PFA-KL with data from different target domains. Note how PFA exploits the knowledge of the target domain (b), creating recipes adapted to the task complexity.

- **PFA fine**: Train from scratch on domain $\mathcal{D}_A$, run PFA-KL on domain $\mathcal{D}_Z$ and train compressed architecture using filter selection on $\mathcal{D}_Z$.

The results in Fig. 3 show how the *PFA fine* strategy performs very similarly to the full fined tuned model (*Full fine*), while obtaining models more than 4 times smaller. Moreover, The *PFA fine* strategy largely outperforms the full model trained from scratch on the target domain (*Full scratch*).

The PFA-KL recipes when adapting from $\mathcal{D}_A^{C10}$ are very similar to each other, independently of the target doamin (Fig. 3d), while those obtained adapting from $\mathcal{D}_A^{C100}$ vary (Fig. 3b). We believe that the fact that the target domain is a subset of $\mathcal{D}_A^{C100}$ and *not* of $\mathcal{D}_A^{C10}$ is the explanation to this effect. In the $\mathcal{D}_A^{C100}$ case responses are stronger and selective, since the images used for PFA were also used for training, leading to stronger correlations. PFA is able to exploit such effect, obtaining recipes adapted to the complexity of the task. Note how PFA obtains recipes with more filters for the 10 class subsets (Fig. 3b). On the other hand, the model trained on $\mathcal{D}_A^{C10}$ has never seen the target domain, therefore this generates less correlation. As a result PFA generates more uniform recipes, as shown in Fig. 3d.

These results show how PFA is able to transfer knowledge to different domains. When the original network had been exposed to the target domain (Figs. 3a, 3b), PFA is able to exploit such knowledge and create recipes adapted to the task. On the other hand, PFA obtains more uniform (and still accurate, Figs. 3c, 3d) recipes when the original network had not been exposed to the target domain.

## 5 CONCLUSIONS

Two effective, and yet easy to implement techniques for the compression of neural networks using Principal Filter Analysis were presented: PFA-En and PFA-KL. These techniques exploit the inherent correlation of filter responses within layers to compress networks without compromising accuracy. These techniques can be applied to the output response of any layer with no knowledge of the training

procedure or the loss function. While easy to implement, both algorithms surpass state of the art results in terms or compression ratio with the advantage that PFA-KL is parameter free and PFA-En has only a single intuitive parameter: the energy to be preserved in each layer or a desired network characteristic (such as a target footprint or FLOPs).

By performing spectral analysis on the responses rather than the weights, PFA allows users to take advantage of transfer learning in order to perform domain adaptation and derive smaller specialized networks that are optimally designed for a specific task, while starting from the same base model.

It is interesting to observe that all compression algorithms in the state of art, including PFA-KL, do not converge if applied repeatedly. This is due to the retraining step which alters the weights in order to optimize an objective function that does not take into account compression criteria. In practical applications this is not a limitation since most of the models tend to be over specified and users can apply compression algorithms until the accuracy falls below an acceptable level. However, it is an interesting theoretical limitation that could inspire future work. Specifically for PFA, this could be a strategy that can also suggest expanding layers so that an optimal size could be reached at convergence.

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

# Appendices

## A  SIMPLECNN ARCHITECTURE

SimpleCNN is composed of the following layers: 3 conv layers of size 96x3x3, a drop-out layer, 3 conv layers of size 192x3x3, drop-out layer, 1 conv layer of size 192x3x3, 1 conv layer of size 192x1x1, 1 conv layer of size [number of classes]x1x1, and finally an average pooling before the softmax layer. We use batchnorm and ReLU activations after every convolutional layer. The drop-out probability is set to 0.5.

## B  RESNET-56 RESULTS ON CIFAR-10 AND CIFAR-100

Results of PFA applied to Resnet-56 on CIFAR-10 and CIFARO-100 are shown in B.1a and B.1b

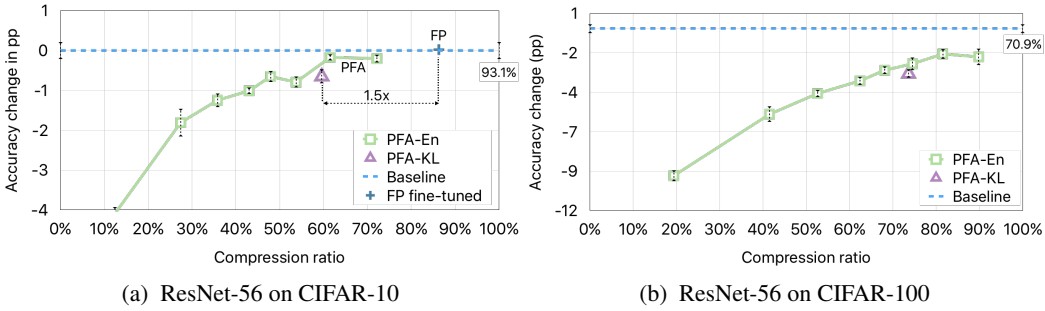

(a) ResNet-56 on CIFAR-10

(b) ResNet-56 on CIFAR-100

Figure B.1: Results for ResNet-56 compressed networks. Accuracy change in the $y$ axis is reported in percentage points. Notice how the accuracy obtained by PFA is comparable to the state of the art while achieving much better compression ratios.

