# OpenReview forum: "NETWORK COMPRESSION USING CORRELATION ANALYSIS OF LAYER RESPONSES"
_ICLR.cc/2019/Conference_

### Official Review · AnonReviewer3 · 2018-10-30
**Interesting hyper-parameter free compression, but missing experiments and clarification/corrections needed**

**Rating:** 5
**Confidence:** 4

**Review:**

This paper introduces an approach to compressing a trained neural network by looking at the correlation of the filter responses in each layer. Two strategies are proposed: one based on trying to preserve the energy of the original activations and one based on looking at the KL divergence between the normalized eigenvalues of the activation covariance matrix and the uniform distribution.

Strengths:
- The KL-divergence-based method is novel and has the advantage of not requiring to define any hyper-parameter.
- The results show the good behavior of the approach.

Weaknesses:

Method:
- One thing that bothers me is the spatial max pooling of the activations of convolutional layers. This means that is two filters have high responses on different regions of the input image, they will be treated as correlated. I do not understand the intuition behind this.
- In Section 2, the authors mention that other methods have also proposed to take the activation into account for pruning, but that they aim to minimize the reconstruction error of these activations. In fact, this is also what PFA-En does; for a given dimension, PCA gives the representation that minimizes the reconstruction error. Therefore, the connection between this method and previous works is stronger than claimed by the authors.
- While it is good that the KL-divergence-based method does not rely on any hyper-parameter, the function \psi used in Eq. 3 seems quite ad hoc. As such, there has also been some manual tuning of the method.

Experiments:
- In Table 1, there seems to be a confusion regarding how the results of FGA are reported. First, in (Peng et al., 2018), the %FLOPS is reported the other way around, i.e., the higher the better, whereas here the lower the better. Similarly, in (Peng et al., 2018), a negative \Delta in accuracy means an improved performance (as stated in the caption of their Table 2, where the numbers reported here were taken). As such, the numbers reported here, and directly taken from this work, are misinterpreted.
- Furthermore, Peng et al., 2018 report much better compression results, with %FLOP compression going up to 88.58%. Why are these results not reported here? (To avoid any misunderstanding, I would like to mention that I am NOT an author of (Peng et al., 2018)).
- Many of the entries in Table 1 are empty due to the baselines not reporting results on these datasets or with the same network. This makes an actual comparison more difficult.
- Many compression methods report results on ImageNet. This would make this paper more convincing.
- While I appreciate the domain adaptation experiments, it would be nice to see a comparison with Masana et al., 2017, which also considers the problem of domain adaptation with network compression and, as mentioned in Section 2, also makes use of the activations to achieve compression.

Related work:
- It is not entirely clear to me why tensor factorization methods are considered being so different from the proposed approach. In essence, they also perform structured network pruning.
- The authors argue that performing compression after having trained the model is beneficial. This is in contrast with what was shown by Alvarez & Salzmann, NIPS 2017, where incorporating a low-rank prior during training led to higher compression rates.
- The authors list (Dai et al., 2018) as one of the methods that aim to minimize the reconstruction error of the activations. Dai et al., 2018 rely on the mutual information between the activations in different layers to perform compression. It is not entirely clear to me how this relates to reconstruction error.

Summary:
I do appreciate the idea of aiming for a hyper-parameter-free compression method. However, I feel that there are too many points to be corrected or clarified and too many missing experiments for this paper to be accepted to ICLR.

After Response:
I appreciate the authors' response, which clarified several of my concerns. I would rate this paper as borderline. My main concern now is that the current comparison with existing method still seems too incomplete, especially with ResNet architectures, to really draw conclusions. I would therefore encourage the authors to revise their paper and re-submit it to an upcoming venue.

---

> ### Author Response · Authors · 2018-11-09
> **Answers to comments on weaknesses**
>
> Thank you for your comments. We appreciate your review.
>
> > ... spatial max pooling... I do not understand the intuition behind this.
>
> Pooling is a relaxation to ease the next step in the process. Jordao et al. 2018 compares different forms of pooling for compression: global max pooling (as in PFA), avg pooling and a spatial preserving max pooling. They observe that global max pooling performs the best.
>
> The intuition is that if two filters are correlated they might be redundant for the end task, even if they learn different features. For example, in order to decide if an image contains a face there is no need to detect nose, mouth,  eyes, etc... one (or more) of these features might be sufficient. That said, we agree that exploring alternatives to max pooling is a potential for future research.
>
> > ... other methods have also proposed to take the activation into account for pruning, ... but they aim to minimize the reconstruction error.... In fact, this is also what PFA-En does;
>
> PFA-En uses the spectral energy of the filters' responses only to decide how many filters should be preserved. Our filter selection does not account for the reconstruction error.
>
> > While it is good that the KL-divergence-based method does not rely on any hyper-parameter, the function \psi used in Eq. 3 seems quite ad hoc. As such, there has also been some manual tuning of the method.
>
> \psi could be tuned to a given task, though we do not. The proposed \psi is the function that empirically worked the best in an initial evaluation, and has not been tuned in any of our experiments.
>
> > In Table 1, there seems to be a confusion regarding how the results of FGA are reported. ...
>
> Thank you for spotting this mistake.
>
> > Peng et al., 2018 report much better compression results, with %FLOP compression going up to 88.58%. Why are these results not reported here?
>
> Table 1 is meant to provide a numerical comparison for similar compression rates. Fig. 2b provides all compression rates for FGA (Peng et al.). However, we made the same mistake in reporting the numbers for FGA. Here is the correct comparison:
>      Footprint; Accuracy change
> FGA  11.31%;   -1.93%
> PFA   9.64%;    -2.37%
>
> FGA  13.20%;    -0.57%
> PFA  14.37%%;  -0.27%
>
> FGA  18.54%;    -0.02%
> PFA  19.27%;    +0.50%
>
> FGA  39.67%;    +0.39%
> PFA  43.11%;    +1.70%
>
> > Many of the entries in Table 1 are empty ... This makes an actual comparison more difficult.
>
> We agree. Sadly there is no agreed benchmark. We hope, that the amount of comparison provided is enough for the reader to form an opinion. Such opinion should also be influenced by other factors (beyond numbers): easy of implementation, practicality, ease of parameters tuning, etc...
>
> > Many compression methods report results on ImageNet. This would make this paper more convincing.
>
> We have just completed the experiments on ImageNet. Here is a summary:
>                                                         Footprint;  Top1 change; Top5 change
> PFA-KL from scratch                   69.30%	    -1.89%             -0.97%
> PFA-KL with filter selection        69.30%	    +2.39%            +1.41%
>
> > While I appreciate the domain adaptation experiments, it would be nice to see a comparison with Masana et al., 2017.
>
> Our results on domain adaptation are meant to support our claim that with PFA different complexities in the tasks lead to different compressions.
>
> > It is not entirely clear to me why tensor factorization methods are considered being so different from the proposed approach.
>
> In the paper, by “tensor factorization” we refer to those algorithms that split weight tensor into a sequence of smaller tensors. By “structured pruning” we refer to those algorithms (like PFA) that remove full filters from the current layer, without attempting to replace or approximate them. This terminology follows other discussions, such as Peng et al. 2018, Liu et al. 2017, and Li et al. 2017.
>
> > The authors argue that performing compression after having trained the model is beneficial. This is in contrast with what was shown by Alvarez&Salzmann, NIPS 2017.
>
> Alvarez&Salzmann (A&S) claim that is beneficial to modify the original loss in order to induce some properties in the full model in order to ease compression. This is not in contrast with our findings. The difference between A&S and PFA is that PFA does not need to modify the loss. From here on, the workflow is the same: we both compress after training and fine-tune the compressed model.
>
> > I do appreciate the idea of aiming for a hyper-parameter-free compression method. However, I feel that there are too many points to be corrected or clarified and too many missing experiments for this paper to be accepted to ICLR.
>
> Thank you for your time and consideration; we hope you will find our answers and new experiments satisfactory. We believe our work is stronger after addressing your concerns. Please let us know if you have further questions and consider updating your rating.

---

> > ### Comment · AnonReviewer3 · 2018-12-02
> > **A few more comments**
> >
> > Thank you for the clarification and for the new results. I nonetheless have a few more comments/questions:
> >
> > - Can you provide the full reference to Jordao et al., 2018? It does not seem to be cited in the paper.
> >
> > - Peng et al., 2018 report results in terms of FLOP compression. For completeness of the comparison, it would be interesting to also have these numbers for the detailed results you provided above.
> >
> > - Regarding Table 1, you mentioned that there is no agreed benchmark. While I agree, several papers report results using a ResNet-34. Your choice of a ResNet-56, for which there are almost no other results, seems strange.

---

> > > ### Author Response · Authors · 2018-12-03
> > > **Further clarifications**
> > >
> > > Thank you for your further feedback.
> > >
> > > > Can you provide the full reference to Jordao et al., 2018?
> > >
> > > This is the full title and link: “Pruning Deep Neural Networks using Partial Least Squares” (https://arxiv.org/pdf/1810.07610v1.pdf). We will add this reference to our paper too.
> > >
> > > > Peng et al., 2018 report results in terms of FLOP compression. For completeness of the comparison, it would be interesting to also have these numbers for the detailed results you provided above
> > >
> > > The information about FLOPs is already available in Table 1. In summary: Peng's algorithm (FGA) achieves a better FLOPs compression than our proposal (PFA), while PFA achieves a better footprint reduction and accuracy improvement  than FGA.
> > >
> > > Specifically, the FLOPs of the compressed model with respect to the original architecture are 43% for FGA  (i.e. 57% reduction) and 52.7% for PFA (i.e., 47.3% reduction). On the other hand, the footprint of the compressed model with respect to the original architecture is 74.4% for FGA (i.e. 25.6% reduction) and  69.3% for PFA (i.e., 30.7% reduction). Finally the Top-1 and Top-5 accuracy improvement with respect to the original model are 0.82% and 0.94% respectively for FGA, and 2.39% and 1.41% respectively for PFA.
> > >
> > > > While I agree, several papers report results using a ResNet-34. Your choice of a ResNet-56, for which there are almost no other results, seems strange.
> > >
> > > Thank you for pointing out this aspect. We have provided results on ResNet-56 because we followed the experimental settings of Li et al. for CIFAR-10. In retrospect we agree that ResNet-34 would have been a more popular choice. We do not expect the conclusions about PFA to change with ResNet-34. We believe we have provided enough quantitative and qualitative information for a reader to form her or his own judgment  about how PFA compares with the state of art. We are, however, happy to do those experiments and report the new results in the paper if this helps to remove any doubts.
> > >
> > > Thank you again for your review, we appreciate the time you invested in it.

---

> > > ### Author Response · Authors · 2018-12-04
> > > **ResNet-34 vs ResNet-56**
> > >
> > > Dear reviewer,
> > >
> > > After exploring the experimental settings of the different state of the art techniques used in our comparison, we see ResNet-34 is not typically used with CIFAR, so unfortunately adding ResNet-34 experiments will not provide additional comparisons.
> > >
> > > Almost all papers present results on CIFAR solely with VGG-16. Only us and Li et al. additionally report results with ResNet (and we both use ResNet-56).
> > >
> > > Papers that use ResNet-34 do so mainly for ImageNet and often include VGG-16 (which we have for comparison).
> > >
> > > In order to augment Table 1 (as you suggest), we will include results for ResNet-34 on ImageNet.
> > >
> > > Thank you once more.

---

### Official Review · AnonReviewer2 · 2018-11-02
**A decent pruning strategy.**

**Rating:** 6
**Confidence:** 3

**Review:**

The paper proposes to prune convolutional networks by analyzing the observed correlation between the filters of a same layer as expressed by the eigenvalue spectrum of their covariance matrix. The authors propose two strategies to decide of a compression level, one based on an eigenvalue threshold, the other one based on a heuristic based on the KL divergence between the observed eigenvalue distribution and the uniform one. This is a bit bizarre but does not require searching a parameter. Once one has decided the number of filters to keep, one can either retrain the network from scratch, or iteratively remove the most correlated filters, which, unsurprisingly, work better.

The authors perform credible experiments on CIFAR-10 and CIFAR-100 that show the results one would expect. They should probably have run ImageNet experiments because many earlier papers on this topic use it as a benchmark and because the ImageNet size often reveals different behaviors.

In conclusion, this is a very decent paper, but not a very exciting one.

-------------

After reading the authors' response and their additional experiments, I still see this work as a very decent paper, but not a very exciting one. This is why I rank this paper somewhat above the acceptance threshold. I could be proven wrong if this approach becomes the method of choice to prune networks, but I would need to see a lot more comparisons to be convinced.

---

> ### Author Response · Authors · 2018-11-09
> **Preview of the results on ImageNet**
>
> Thank you for your positive comments and for your feedback and time. Please find a preview of the results on ImageNet below.
>
> > They should probably have run ImageNet experiments because many earlier papers on this topic use it as a benchmark and because the ImageNet size often reveals different behaviors.
>
> We agree with the reviewer. We have just completed the experiments on ImageNet. Here is a summary of the results.
>
>                                                         Footprint;  Top1 change; Top5 change
> PFA-KL from scratch                   69.30%	    -1.89%             -0.97%
> PFA-KL with filter selection        69.30%	    +2.39%            +1.41%
>
> We will report these results. In addition to the experiments on ImageNet, are there other enhancements that would elevate the paper?

---

### Official Review · AnonReviewer1 · 2018-11-03
**Interesting approach, second time reviewing**

**Rating:** 5
**Confidence:** 5

**Review:**

This paper proposes a compression method based on spectral analysis. The basic idea is to analyse correlation between responses of difference layers and select those that are more relevant discarding the others. That, in principle (as mentioned in the paper) differs from other compression methods based on compressing the weights independently of the data being used. Therefore, in theory (nothing shown), different task would provide different outputs while similar works would compress in the same manner.

Then, the paper proposes a greedy algorithm to select those filters to be kept rather than transforming the layer (as it has been usually done in the past [Jaderberg et al]). This is interesting (from a practical point of view) as would lead to direct benefits at inference time.

This is the second time i review this paper. I appreciate the improvements from the first submission adding some interesting results.

I still miss results in larger systems including imagenet. How all this approach actually scales with the complexity of the network and the task?

There have been recent approaches incorporating low-rank approximations that would be interesting to couple with this approach. I am surprissed these are not even cited ('Compression aware training' at NIPS 2017 or Coordinating filters at ICCV2017 both with a similar approach (based on weights tho). Pairing with those approaches seems a strong way to improve your results.

---

> ### Author Response · Authors · 2018-11-09
> **Evidence of our claim, and preview of the results on ImageNet (including complexity analysis and actual times)**
>
> We appreciate your review and would like to thank you for your feedback and time. Please find our answers below.
>
> > Therefore, in theory (nothing shown), different task would provide different outputs while similar works would compress in the same manner.
>
> The experiments presented in section 4.5 and Figure 3 provide evidence that with PFA different complexities in the tasks lead to different architectures; similar tasks lead to similar architectures. We will clarify the connection between the claim and Section 4.5. In those experiments, we show how the architecture produced by PFA (starting from the same trained model) differs depending on the complexity of the task. For example, VGG-16 trained on CIFAR-100 and then compressed using 10 labels (R1 line in Figure 3.b) has 150 filters in the last layer (and overall it has 52% of the original filters), whereas a simpler task with 2 labels (S4 line in Figure 3.b) has only 45 filters in the last layer (and overall it has 36% of the original filters). The compression obtained in the two cases is clearly different and reflects the complexity of the tasks with respect to what the trained model has already learnt. On the other hand, the two tasks with 10 labels (R1 and R2 lines) lead to similar compression.
>
> > I still miss results in larger systems including imagenet. How all this approach actually scales with the complexity of the network and the task?
>
> We have just completed the experiments on ImageNet. Here is a summary. In terms of complexity, we have a dedicated section in Appendix C. However, we will consider move the main conclusions to the main paper. In summary, the complexity of the PFA algorithm per layer is O(mn^2 + n^3), where n is the number of filters, and m is the number of samples. The task does not affect the complexity because the labels are not used in PFA. For the experiment on ImageNet (m=1.2M, ILSVRC2012) PFA took the longest on the last two layers (n = 4096) which were computed in roughly 120 seconds. Here is the full table of times when computing PFA sequentially (non-parallel CPU implementation):
>
> block 0 (64 filters)
> conv0: 1.19s
> conv1: 1.28s
>
> block 1 (128 filters)
> conv0: 2.74s
> conv1: 2.83s
>
> block 2 (256 filters)
> conv0: 4.26s
> conv1: 4.83s
> conv2: 4.78s
>
> block 3 (512 filters)
> conv0: 8.92s
> conv1: 9.43s
> conv2: 9.92s
>
> block 4 (512 filters)
> conv0: 9.18s
> conv1: 9.22s
> conv2: 9.22s
>
> fully connected block (4096 filters)
> fc1: 142.17s
> fc2: 112.74s
>
> As mentioned in Appendix C, PFA has to run once at the end of the training step and as shown above the time consumed by PFA is a negligible compared to the whole training time. In exchange for this marginal extra-time PFA provides the long-term benefit of a smaller footprint and faster inference, which in the lifetime of a deployed network will quickly surpass the time initially required by PFA. In addition, when working in a setting where the network is periodically re-trained with new incoming data, having a smaller network will add to the saved time.
>
> In terms of performance, these are the results of PFA-KL on ImageNet
>                                                         Footprint;  Top1 change; Top5 change
> PFA-KL from scratch                   69.30%	    -1.89%             -0.97%
> PFA-KL with filter selection        69.30%	    +2.39%            +1.41%
>
> > There have been recent approaches incorporating low-rank approximations that would be interesting to couple with this approach. I am surprissed these are not even cited ('Compression aware training' at NIPS 2017 or Coordinating filters at ICCV2017 both with a similar approach (based on weights tho). Pairing with those approaches seems a strong way to improve your results.
>
> Thank you for the references. We will add them to our state-of-the-art review. Both techniques are interesting and could further improve the compression rate of PFA. The approach of modifying the original loss in order to induce a specific property in the full model is smart but goes against the philosophy that we have adopted for PFA. We envision PFA to be easy to use: no-parameter strategy (PFA-KL), no need to modify the original loss function (which would require additional hyper-parameters tuning), and the ability to start from pre-trained models or to use known training hyper-parameters. Nevertheless we are eager to consider how these other techniques could be paired with PFA in future work and see what level of improvement can be achieved.
>
> We hope we have answered all your concerns. Please let us know if you think there are more opportunities for improvement.

---

### Author Response · Authors · 2018-11-15
**New version of the manuscript available**

We have uploaded a new version of our paper to address reviewers' comments. Here are the highlights of the changes:

- We have added the references mentioned by the reviewers.
- We have clarified a few sentences to avoid misunderstandings.
- We have included a section regarding the complexity analysis of PFA, as well as the actual time required by PFA to run on each layer of VGG-16 using the ImageNet dataset.
- We have included compression results on the ImageNet dataset.
- We have corrected the numbers in Table 1 regarding the FGA algorithm.

Thanks to the changes recommended by the reviewers, the claims in the paper are strengthened.  Independently of the architecture or dataset, PFA consistently provides better compression or accuracy than the state of the art.

We would like to thank the reviewers once more for their valuable feedback. We hope they will find the changes satisfactory or we will wait for new feedback.

---

### Author Response · Authors · 2019-02-11
**Closing comments**

Thanks to the reviewers and the AC for their feedback. This process of review has strengthened and clarified our paper significantly.  Closing comments about the final review topics can be found below:

1. "Results on large-scale tasks such as Imagenet" was addressed in the rebuttal, as acknowledge by the AC.

2.  "Compression after the fact may not be as good as training with a modified loss function that does compression jointly”.  Without discussing here the merits and the drawbacks of changing the loss, in our comparisons, we showed that we obtain better results than other state of the art techniques, two of which do modify the loss function. There may be in the future techniques which effectively use modification of the loss, however, to the best of our knowledge, this statement is currently not supported by evidence.

3.  "Insufficient comparisons on ResNet architectures which make comparisons against previous works harder”. The simplicity and improvements of our proposed technique are evident in the experiments we have presented. Experiments show that our technique achieves better results on a variety of architectures (a simple CNN, VGG-16 and ResNet) and datasets (CIFAR10, CIFAR100 and ImageNet).  ResNet on ImageNet (we used VGG-16 instead) is the only missing combination. As we expand our presented results, we look forward to covering all experiments which reviewers feel are critical.  As it is always possible to describe a subjectively useful or missing experiment, we hope the community will continue to revise evaluations criteria to emphasize progress rather than coverage.

---

### Meta-Review · Area_Chair1 · 2018-12-14
**Interesting approach to compression based on analyzing filter activations.**

**Confidence:** 4
**Recommendation:** Reject

**Metareview:**

The authors propose a technique for compressing neural networks by examining the correlations between filter responses, by removing filters which are highly correlated. This differentiates the authors’ work from many other works which compress the weights independent of the task/domain.

Strengths:
Clearly written paper
PFA-KL does not require additional hyperparameter tuning (apart from those implicit in choosing \psi)
Experiments demonstrate that the number of filters determined by the algorithm scale with complexity of the task

Weaknesses:
Results on large-scale tasks such as Imagenet (subsequently added by the authors during the rebuttal period)
Compression after the fact may not be as good as training with a modified loss function that does compression jointly
Insufficient comparisons on ResNet architectures which make comparisons against previous works harder

Overall, the reviewers were in agreement that this work (particularly, the revised version) was close to the acceptance threshold. In the ACs view, the authors addressed many of the concerns raised by the reviewers in the revisions. However, after much deliberation, the AC decided that the weaknesses 2, and 3 above were significant, and that these should be addressed in a subsequent submission.